# Stochastic simulation and statistical inference platform for visualization and estimation of transcriptional kinetics

Gennady Gorin[1], Mengyu Wang[2,3], Ido Golding[2,3], Heng Xu[4,5]*

**1** Division of Chemistry and Chemical Engineering, California Institute of Technology, Pasadena, California, United States of America, **2** Department of Physics, Grainger College of Engineering, University of Illinois at Urbana-Champaign, Urbana, Illinois, United States of America, **3** Center for the Physics of Living Cells, University of Illinois at Urbana-Champaign, Urbana, Illinois, United States of America, **4** School of Physics and Astronomy, Shanghai Jiao Tong University, Minhang District, Shanghai, China, **5** Institute of Natural Sciences, Shanghai Jiao Tong University, Minhang District, Shanghai, China

* heng_xu@sjtu.edu.cn

**Data Availability Statement:** All relevant data are within the manuscript and its Supporting Information files.

## Abstract

Recent advances in single-molecule fluorescent imaging have enabled quantitative measurements of transcription at a single gene copy, yet an accurate understanding of transcriptional kinetics is still lacking due to the difficulty of solving detailed biophysical models. Here we introduce a stochastic simulation and statistical inference platform for modeling detailed transcriptional kinetics in prokaryotic systems, which has not been solved analytically. The model includes stochastic two-state gene activation, mRNA synthesis initiation and step-wise elongation, release to the cytoplasm, and stepwise co-transcriptional degradation. Using the Gillespie algorithm, the platform simulates nascent and mature mRNA kinetics of a single gene copy and predicts fluorescent signals measurable by time-lapse single-cell mRNA imaging, for different experimental conditions. To approach the inverse problem of estimating the kinetic parameters of the model from experimental data, we develop a heuristic optimization method based on the genetic algorithm and the empirical distribution of mRNA generated by simulation. As a demonstration, we show that the optimization algorithm can successfully recover the transcriptional kinetics of simulated and experimental gene expression data. The platform is available as a MATLAB software package at https://data.caltech.edu/records/1287.

## Introduction

Transcription has been the focus of intensive study due to its cornerstone role in cell activity regulation. Recent advances in fluorescent imaging have enabled mRNA detection at single-molecule resolution in individual cells, in both live and fixed samples [1,2]. Spatial analysis of mRNA signals allows the identification [3,4] and quantification [5] of nascent (actively transcribed) mRNA, which offers a direct window into the kinetics of gene transcription, with minimal interference from downstream effects [5], at the level of a single gene copy [6].

Converting high-resolution experimental data into theoretical understanding of transcription requires simultaneous modeling of both nascent and mature species of mRNA.

**Funding:** The authors were funded by the following sources during the completion of this research: GG: NIH U19MH114830. National Institutes of Health. nih.gov. GG: Undergraduate Asian Studies Internship Award (U-ASIA) 2017. Rice University Chao Center for Asian Studies. chaocenter.rice.edu. MW, IG: R01 GM082837. National Institutes of Health. nih.gov. MW, IG: PHY 1430124. National Science Foundation. nsf.gov. HX: 2018YFC0310803. National Key R&D Program of China. http://most.gov.cn/ HX: 11774225. National Natural Science Foundation of China. nsfc.gov.cn. HX: Thousand Talents Plan of China, Program for Young Professionals. 1000plan.org.cn. HX: 18ZR1419800. National Science Foundation of Shanghai. stcsm.sh.gov.cn. HX: 1013907. Burroughs Wellcome Fund Career Award at the Scientific Interface. bwfund.org. The funders had no role in study design, data collection and analysis, decision to publish, or preparation of the manuscript.

**Competing interests:** The authors have declared that no competing interests exist.

Particularly, since at any given moment an mRNA molecule may be in a partially transcribed and/or degraded state, a good model should be able to capture the submolecular features of mRNA. However, current computational models of transcription present challenges for integration with the new wealth of microscopy data. Most models do not distinguish between nascent and mature mRNA or model the transcript length [7–11]. As recently noted [5], several mechanistic models do describe the elongation of nascent mRNA, but do not consider the mature mRNA population and require additional processing for comparison to microscopy data [4,12–14]. Further, studies using these models tend to predict low-order statistics [7,13], which paint a limited picture at biologically low molecule numbers [4,15]. Recent methods based on directly solving the chemical master equation (CME), using the finite state projection (FSP) algorithm, yield distributions of the number of molecules [5,15,16]. However, integrating the discrete CME with submolecular features of mRNA is nontrivial, and has only recently been accomplished on a model with a deterministic elongation process [5]. A stochastic stepwise model of transcription, more faithful to the mechanistic details, is not currently tractable using FSP [5] due to exponential growth in the size of the state space with increasing resolution.

Here we present a stochastic simulation platform that aims to capture the complexities of RNA processing. The platform consists of a submolecular implementation of the Gillespie algorithm [17], simulating the gene switching, transcription, and degradation expected in a prokaryotic system. Transcription and degradation occur in a stochastic fashion, where the initiation and individual steps of elongation are Poisson processes. The algorithm outputs time-dependent fluorescent probe signals, calculated from the overlap of intact RNA and probe-covered regions. The probe signals are provided as cell-specific readouts and as aggregated histograms, mimicking live-cell (MS2) and fixed-cell (smFISH) fluorescence data, respectively [1,2]. Using a GUI, a user can input simulation parameters and examine time-dependent statistics, as well as animate the instantaneous molecule states.

We use the platform to approach the inverse problem of biological parameter estimation. A recent investigation demonstrated that entire distributions are required to reliably estimate parameter values from single-cell mRNA data [15]. To perform parameter estimation based on these empirical distributions, we implement a heuristic approach based on iteratively minimizing mean squared errors and Wasserstein distances of different observables [18]. This approach represents a novel method of estimating plausible regions for multiple parameters using time-series data with multiple observables, without making assumptions regarding the functional form of the distributions. Thus, the platform provides a flexible simulation environment to implement reaction mechanisms as well as a search algorithm designed to directly test those mechanisms' parameters against experimental data. The GUI and search algorithm are available at https://data.caltech.edu/records/1287.

## Results

### Model and simulation platform

Our platform models a common formalism for the mRNA transcription process [5,7], with a series of stochastic reactions, including promoter turn-on and turn-off, transcription initiation, elongation, RNase (ribonuclease) binding, and degradation (S1 File of S1 Table). Specifically, promoter activity is represented as a two-state switch. In the active ("on") state, transcription can be initiated. The nascent mRNA strand elongates from the 5' to the 3' end, in a series of discrete steps. Upon reaching the end of the template gene, the mature mRNA molecule is released from the gene. Regardless of RNA maturity, RNase can bind to the 5' end of the mRNA, causing the strand to begin stepwise degradation at an average rate assumed to be identical to the elongation speed [19]. The process is depicted in Fig 1A. The physiology of the transcribed

gene is parametrized by the turn-on rate $k_{on}$, the turn-off rate $k_{off}$, the transcription initiation rate $k_{ini}$, the degradation initiation rate $k_{deg}$, the elongation speed $v_{el}$, and the gene length $L$. The experimental parameters include the timespan of the experiment $T_{end}$, as well as the probe span vector $(P_5, P_3)$ defining its 5' and 3' limits of coverage with respect to the length of the gene [5], as shown in Fig 1A.

The platform performs stochastic simulation of the model using the Gillespie algorithm [17,20], then estimates the fluorescence of each mRNA molecule from the size of its region targeted by fluorescent probes. Specifically, we simulate the production and degradation of each mRNA molecule in the cell, whose status can be defined by four variables, i.e. two integers that define 5'- and 3'-most nucleotides of the transcript and two Boolean variables that define whether the mRNA is polymerase-bound (nascent) and/or RNase-bound (degrading). The gene state (on or off) is defined by a single Boolean variable. To convert the simulated mRNA molecule ensemble (Fig 1B) to the experimentally observed fluorescent signal, we calculate the overlap between the intact RNA and the probe coverage (single realization shown in Fig 1C); the probe readout is rescaled to molecule number using the fluorescence of a single intact molecule [16]. The resolution of the simulation is determined by the number of cells and the number of steps taken to fully elongate or degrade each molecule.

Model simulation is implemented in MATLAB 2018a [21]. A simple graphical user interface (GUI), provided as a MATLAB app at https://data.caltech.edu/records/1287, runs the simulation for a user-defined parameter set defining the physical parameters and simulation precision. Upon completion, the GUI outputs the time-dependent mean probe signal (in units of molecule number), Fano factor, and instantaneous nascent and total mRNA probe signal histograms, all calculated over the cell population. The mRNA nucleotide spans are used to visualize and animate the transcriptional activity taking place at an individual gene copy (analogous to Fig 1B and 1C; example visualization given in S1 Movie). Our software allows direct

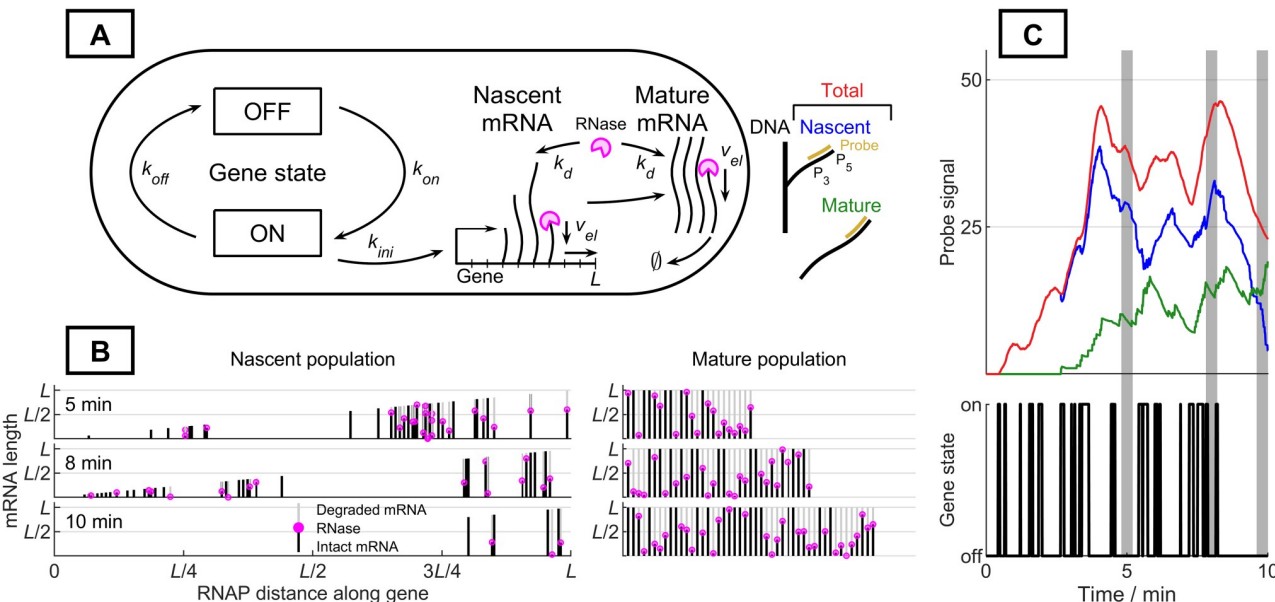

**Fig 1. Model and simulation platform. A**: Model schematic and probe parameterization (gold: probe coverage, $P_3$: 3'-most edge of the probe, $P_5$: 5'-most edge of the probe) **B**: Time-dependent molecule-level visualizations available through the GUI. Trajectory generated using $k_{ini}$ = 100 min⁻¹, $k_{on}$ = 3 min⁻¹, $k_{off}$ = 10 min⁻¹, $k_{deg}$ = 0.5 min⁻¹, $v_{el}$ = 41.5 nt s⁻¹, $T_{end}$ = 10 min, $L$ = 5300 nt, 241 steps of elongation to complete transcription (dark line: intact RNA stretches, light line: degraded RNA stretches, pink circle: RNase molecule). **C**: Single-cell trajectory with simulated nascent and mature fluorescent signals. Parameters same as in **B** (red: total signal, blue: nascent signal, green: mature signal, shaded regions: times displayed in **B**).

simulation of complex experimental designs. For instance, to mimic the commonly-used induction experiment (e.g. the addition of isopropyl β-D-1-thiogalactopyranoside, an inducer of the *lac* promoter, to *E. coli* cells [6]), the simulation starts with no mRNA and undergoes a step increase in the gene turn-on rate. Similarly, to mimic a repression experiment (e.g. the addition of 2-nitrophenyl-β-D-fucoside to *E. coli*), the system starts with a steady-state population of mRNA and undergoes a step decrease in gene turn-on rate [22]. For physiologically plausible transcription in short, infrequent bursts [23], the decrease in $k_{on}$ can also model repression by a step decrease in initiation [6] caused by the addition of rifampicin [24].

## Parameter estimation

Given single-cell time-series fluorescence data that describes nascent and mature mRNA, we seek to estimate the underlying model parameters. We would like to approach this inverse problem by simulating mRNA number distributions for the experimentally available time-points, evaluating an error metric that maps the divergence between the target distribution and each trial distribution to a single number, then minimizing this error by using it as an objective function.

Since metrics based on noisy empirical stochastic distributions do not meet the smoothness assumptions of gradient-based optimizations methods [25], we select a genetic algorithm for optimization. We use the MATLAB implementation of the genetic algorithm [21,26] to sample and evolve points in a parameter space spanning several orders of magnitude for each variable. Consistent with previous investigations, we use a logarithmic parameter search space [15]. Each trial parameter vector {$k_{ini}$, $k_{on}$, $k_{off}$, $k_{deg}$, $v_{el}$} is evaluated using an ensemble of hundreds to thousands of simulated cells. Due to the high computational load (millions of cell trajectories) of a single search, we vectorize the computation and parallelize it across processors on the Amazon Web Services (AWS) cloud [27]. Since cells are independent, the algorithm scales well by parallelization across multiple processors. At the end of the simulation, the parallelized cell ensemble is reassembled into a single population and the statistics defining the error are computed locally, as shown in Fig 2A. To speed up convergence to consistent parameter sets, our heuristic method uses a variable objective function, with five distinct stages that use different error metrics. Details of the metrics are provided in **Methods**.

To test the algorithm's ability to recover known parameters, we generated synthetic data for the turn-on experiment using the following ground truth parameters: $k_{ini}$ = 95 min$^{-1}$, $k_{on}$ = 1 min$^{-1}$, $k_{off}$ = 10 min$^{-1}$, $k_{deg}$ = 0.5 min$^{-1}$, $v_{el}$ = 41.5 nt s$^{-1}$, $T_{end}$ = 15 min, $L$ = 5300 nt, 10,000 cells, and 15 steps of elongation to complete transcription. The procedure used to convert these rates into reaction propensities is described in the S1 File. Relatively coarse simulation quality was used as a proof of concept. The simulations were parallelized across 90 AWS processors. The process of parameter identification is visualized in Fig 2B. We found that the one-sigma interval around the mean estimate included the ground truth parameters (Fig 2C). The convergence of $k_{on}$, $k_{deg}$, and $v_{el}$ throughout the search is relatively well-behaved and close to monotonic; however, $k_{off}$ and $k_{ini}$ are far more challenging to estimate (Fig 2D).

We compare the mean signals of nascent and total RNA simulated using the one-sigma estimate interval (Fig 2E), as well as the corresponding distributions simulated using the mean estimate (Fig 2F), to the synthetic ground truth data. Comparison at both levels demonstrates convergence. To cross-validate the search, we compare repression simulations generated from the ground truth and estimated parameters. The nascent and total means are consistent (Fig 2G). To test the robustness of the fitting algorithm, we apply the search procedure to the turn-on data generated using a range of $k_{on}$ and $k_{off}$ values, mimicking the regulatory parameter modulation hypothesized to occur *in vivo* [28]. The results suggest consistent performance

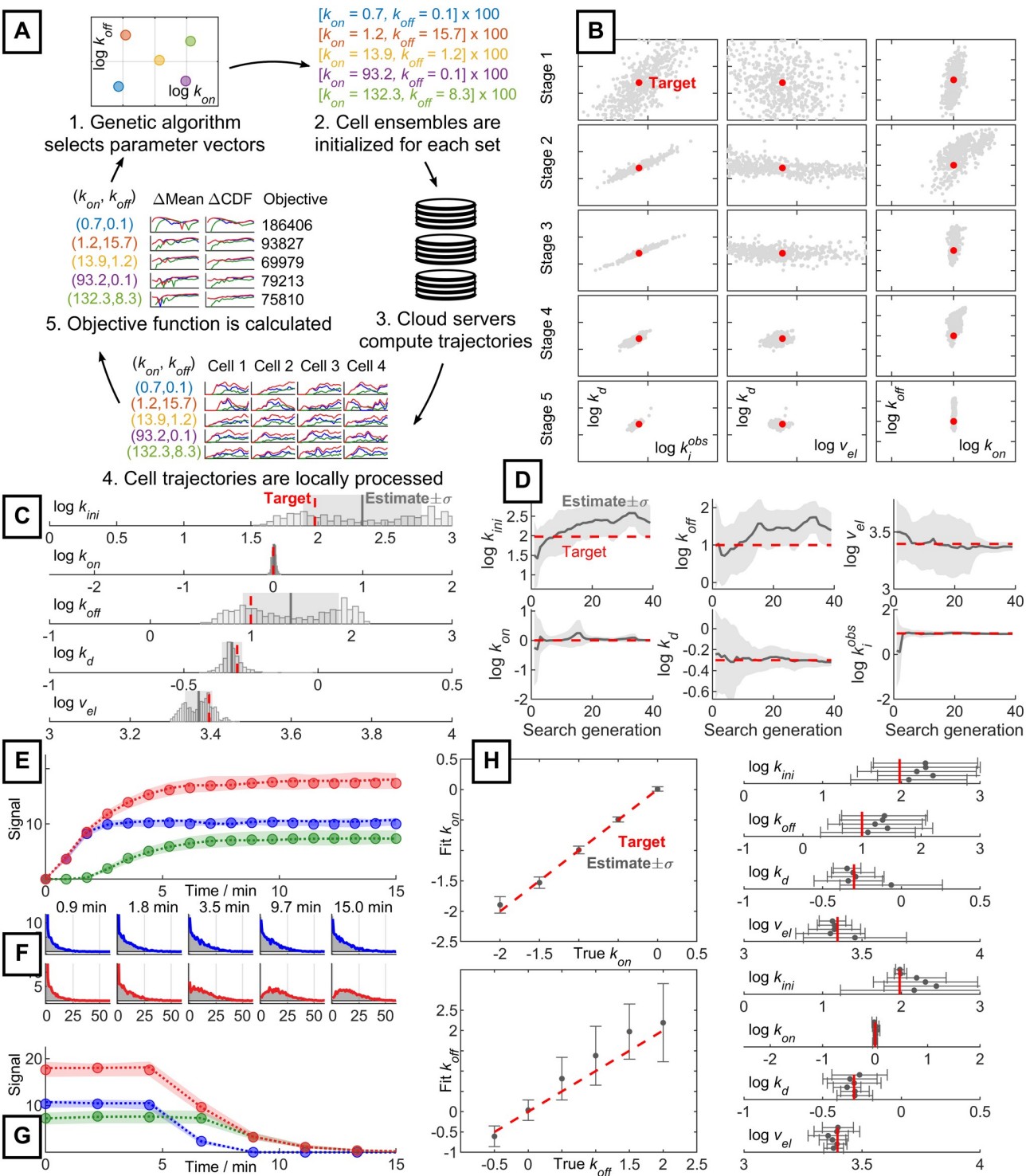

**Fig 2. Parameter estimation process and performance. A**: Parallelized calculation of the search objective function for a set of trial parameters (ΔMean: mean squared error, ΔCDF: Wasserstein distance, Objective: error function value). **B**: Convergence of the genetic algorithm at the end of each stage of the search (red: ground truth target, gray: population of parameter estimates). **C**: Final trial parameter population from **B** (red: ground truth target, histogram: estimate population, gray line: mean estimate, gray region: one-sigma region of estimates). **D**: Evolution of parameter estimates throughout the search process (red: ground truth target, gray line: mean estimate, gray region: one-sigma region of estimates). **E**: Comparison of mean probe signal between target and fit (circles: target data, dotted line: mean parameter estimate, shaded region around dotted line: signal spanned by fifty estimates sampled from the one-sigma region). Colors as in Fig 1. **F**: Comparison of copy-number distributions between target and fit (shaded gray regions: target histogram, colored lines: histogram generated from mean parameter estimate, top row/blue: nascent mRNA distribution, bottom row/red: total mRNA distribution). **G**: Comparison of mean probe signal between target and fit in turn-off cross-validation experiment. Convention as given for **E**. **H**: Estimation of modulated parameters. Top trial modulates $k_{on}$, bottom trial modulates $k_{off}$. All other parameters are constant but unknown to the search algorithm and are fit independently (red: ground truth target, gray dots and error bars: mean estimate and one-sigma region of three replicates).

throughout the parameter space, although identifiability of high $k_{off}$ is poor (Fig 2H). Encouragingly, all one-sigma intervals include the ground truth parameters.

For additional validation, we ran the search algorithm using synthetic data generated from random parameter vectors, as well as experimental data from a recent study [6]. These procedures are described in the **Further Validation** section of S1 File. We found that the fits successfully reproduced time-dependent distributions of probe signals. However, agreement between the inferred parameters and ground truth (or, for experimental data, FSP estimates) was not guaranteed, especially for $k_{ini}$ and $k_{off}$. As in Fig 2, these gaps in performance appear to correspond to non-uniqueness in mapping from the parameter domain to the observable domain [29], and inability of the genetic algorithm to report degenerate results. We suggest that this degeneracy is best identified by running the search algorithm multiple times and examining the resulting distribution of point estimates from the centers of the search populations. We take this approach in Fig 2H.

## Methods

The Gillespie algorithm is adapted from the original description [17] and implemented in the MATLAB programming language [21]. To account for submolecular degrees of freedom, the simulation uses multiple data structures to describe the system state. Specifically, one multidimensional dynamic array holds the 5' and 3' indices of each mRNA (transcript span), another identifies whether it is being transcribed at a particular gene locus or free in the cytoplasm (RNA polymerase attachment), and a third tracks whether it is being degraded (RNase attachment). Smaller, static arrays track the system time, gene state, and number of mRNA and bound RNase molecules. Each reaction either increments or flips Boolean values in the appropriate state arrays. State variables and reactions are outlined in detail in S1 File; the reaction propensity calculations are given in S1 File of S1 Table.

To perform parameter estimation on turn-on synthetic data, we use a heuristic iterative method based on the genetic algorithm [30]. We alternate between optimizing mean signals and entire distributions. The error metric for the mean signal is the mean squared error. Due to the limited support of empirical distributions, the commonplace minimization of Kullback–Leibler divergence between target and test distributions [31] is inappropriate for comparing distributions [25]. Instead, we use the absolute difference between the target and test cumulative distribution functions (CDFs), which tends to be more robust to noise and sparsity [25]; this metric is commonly known as the Wasserstein or earth mover's distance [18]. We aggregate different time points' Wasserstein distances by weighing them using a uniform or exponential function of time, as described in S1 File.

Empirically, the parameter identifiability is far from uniform throughout the simulated time-series, and different metrics provide sensitivity to different parameters. Further, it is computationally prohibitive to simulate entire trajectories at the beginning of the parameter search, when the relevant region of the five-dimensional search space is not yet known. Therefore, we take an *ad hoc* iterative approach, which incrementally narrows the region of parameters consistent with the observed signals. This heuristic approach is chosen for computational convenience and is not guaranteed to the global parameter optimum.

The parameter domain is shown in Fig 2C. We initialize the search using a uniform distribution over the full parameter domain. The first stage identifies the parameter space consistent with the distributions of nascent signals observed throughout the first few time points of the experiment, essentially acting as an order-of-magnitude filter and eliminating computationally expensive edge regions with extremely high or low transcription. This stage uses a population of 5,000 parameter sets and only keeps the top 10% of best variants; based on Fig 2B, it

identifies $k_{on}$ and a degenerate line containing consistent values of $k_d$ and $k_i^{obs} = k_{ini} \frac{k_{on}}{k_{on}+k_{off}}$. The second stage attempts to truncate this space to parameter values consistent with the mean level of total RNA for the entire time series, and identifies tighter bounds for $k_d$ and $k_i^{obs}$. This and all following stages use populations of 500 trial parameters. The third stage refines the estimate to parameter values consistent with the steady state distribution of total mRNA, and yields tighter bounds for $k_{on}$ and $k_{off}$. The fourth stage uses information from the mean level of nascent RNA for the entire time series, and improves bounds for $v_{el}$. Finally, the fifth stage refines the bounds for $k_{on}$ and $k_{off}$ by performing a high-precision optimization using the metric used in stage 1. By penalizing the objective function for deviating beyond a given radius from the previous stage's parameter region, consistency between different error metrics is enforced, as described in S1 File. More detailed data regarding each stage's penalization and precision are provided in S1 File of S2 Table.

## Discussion

Above we describe a new platform for simulating mRNA transcription and degradation on a submolecular level, available at https://data.caltech.edu/records/1287. Its output is directly comparable to single-cell data of nascent and mature mRNA. The output of each simulation is the empirical distribution of signals for each cell at each time point. Therefore, the platform can simulate both live-cell measurements (which identify cell-specific signals over time) and fixed-cell measurements (which yield population statistics) [1,2]. As the platform is based on the stochastic simulation algorithm, it is relatively straightforward to modify the model to incorporate new reactions, chemical species, regulatory pathways, and labeling schema. The software includes single-cell and statistical visualization tools to facilitate general-purpose use without coding. For resource-intensive parameter space exploration, we suggest heuristics to accelerate convergence. The method demonstrates that parameter estimation from a time series of multiple observables is tractable by heuristic likelihood-free methods. The validation we perform suggests that, by using simulations to generate empirical distributions, this approach is more effective to fit experimental signals than traditional methods when no closed-form solutions or approximations are available; further, the visualization capabilities would be useful for the qualitative description and understanding of such complex systems.

Our platform allows numerical solution of detailed transcription model for both nascent and mature mRNA species, whose CME may not be solved exactly. However, since the approach is simulation-based, the steady state of the system needs to be computed asymptotically from a non-steady state, which may be time-consuming. Specifically, simulating and fitting the steady-state and turn-off experiments may be computationally prohibitive if the scales of kinetic rates are substantially different. Alternatively, it may be possible to use analytical solutions [32,33] to approximate an equilibrium distribution; however, this approach is challenging to generalize and the resulting simulation would no longer be exact.

The parameter identification process may be facilitated by parameter constraints from analytical solutions. For example, if the steady-state solution for the total mean is known, the $k_i^{obs}$ and $k_{deg}$ parameters can be fixed for the optimization procedure, reducing the parameter estimation to the simpler problem of optimization in three-dimensional space of $k_{on}$, $k_{off}$, and $v_{el}$, as shown in S2 Movie.

On the other hand, we suggest that five-parameter inference entirely from moments is infeasible at this time. Typically, fitting $n$ parameters requires $n$ moments. For the current system, signal expectations can be computed [6], but expressions for the higher moments are unknown. Even if they were available, the choice of error model for these higher moments is far from clear, especially in the physiologically important regime of low copy numbers.

Furthermore, we anticipate that the value of this heuristic method rests in applications to models with *ad hoc* mechanisms whose physics are challenging to approach analytically.

Even without moment-based analytical constraints, it is possible to use physical considerations to guide the development of optimization metrics. For example, in a Bayesian framework, the Fisher information of the mean total probe signal is high with respect to $k_d$, but low with respect to $v_{el}$. As shown in Fig 2B, stage 2, which optimizes the total mean probe signal, provides a tight bound on $k_d$ but not $v_{el}$; conversely, stage 4, which optimizes the mean nascent probe signal, yields a tight bound on $v_{el}$. For more complex models, exploratory analysis is necessary to determine the coupling between observables and parameters, but the provided heuristics and physical expectations provide a starting point.

The parameter estimation procedure only uses time-dependent histograms: the platform can generate live- and fixed-cell data, but only attempts to fit fixed-cell data. These biochemical distinctions induce methodological differences for parameter inference. Fixed-cell measurements are necessarily destructive, and kinetics may only be inferred from distribution-level data. In contrast, live-cell signals contain additional information regarding the temporal correlation of a given cell. In the current study, we focus on fitting distribution data for two reasons. Firstly, inference from ensembles can be directly implemented using a variety of divergence metrics that make minimal assumptions regarding the form of the data [25]. On the other hand, inference from time-series requires error models for transitions between observed states, which are generally intractable [34]. Secondly, fixed-cell measurements are amenable to high-throughput experiments, can be scaled to the entire transcriptome via multiplexing [35], produce better signal/noise behavior, and do not require genetic modification [36], contributing to their greater popularity [36]. Therefore, we have optimized the parameter estimation method for the most likely current use case of inference from fixed-cell experiments.

Recent advances in live-cell labeling techniques do suggest that the method may become more practical and popular in the future [37,38]. To anticipate this, we propose several approaches to live-cell data inference, motivated by previous efforts. If the dataset is large enough, the fixed-cell procedure may be sufficient, discarding the temporal correlation information altogether [25]. Alternatively, it is possible to iterate through the data points of a time-series, generating an ensemble of transitions, estimating the likelihood of the observed transition based on a kernel, and optimizing the likelihood by varying model parameters. This approach has been useful for relatively small datasets [34,39,40]. However, its application to multimodal time-series is potentially problematic due to the assumption of smoothness, the complexity of developing robust adaptive kernels, and the well-documented problems accompanying kernel density estimation of multivariate data [41]. Further, it presents computational challenges: the different increments are ostensibly independent due to the Markov property, but the non-unique mapping from the underlying Markov states to the observed probe data prevents the independent initialization of each increment. This feature makes it infeasible to parallelize the estimation of transition probabilities over non-overlapping increments. Several recent publications perform likelihood-based inference on hidden Markov models [37,38]. However, rigorously recasting these methods into the context of likelihood-free simulation is challenging, as is their extension to multimodal data. We suggest that the algorithm described in the **Methods** section can be extended to treat time-series data. Such an algorithm may iterate over a single time-series to incrementally shrink to a consistent parameter region. The selection of the region is based on a non-parametric error metric between the target fluorescence and the ensemble distribution for each trial parameter at the end of each interval. Conceptually, this process iteratively identifies parameter values by optimizing for observed transitions, analogously to previous work [40]. Afterward, independent searches over multiple traces may be aggregated to find a single plausible region. Given the computational expense of

current HMM-based methods [38], an adaptive simulation-based approach may present a viable alternative.

Our platform models the activity of individual gene loci in non-compartmentalized prokaryotic cells with the assumption that transcription follows a two-state random telegraph model with time-homogeneous rate parameters, and elongation and degradation are described by multistep Poisson processes. These assumptions may be violated in the following ways:

1. The description of a eukaryotic system may be of interest. The implementation of eukaryotic transcription would require making significant changes to the reaction schema, such as disabling the degradation of nuclear mRNA and adding a kinetic model of a transport process after the release of the newly transcribed mRNA.

2. Multiple gene copies may be present in a cell [6]. It is straightforward to extend the current model to account for this physiology. For example, S3 Movie shows the correlated dynamics at two gene copies, which may only turn on when an underlying Boolean cell state is on.

3. The two-state switching of gene activation/inactivation may be an over-simplified picture of gene activity. In reality, an $N$-state model may be more accurate [15,42,43]. To consider this effect, our simulation-based framework can be easily extended to include more gene states rather than a single Boolean state.

4. The transcription elongation rate may not be constant, whether due to sequence dependence [44] or polymerase congestion [13,14,45]. The implementation of these rules is challenging using the CME framework. Our simulation-based platform can incorporate sequence-dependent rates by adjusting rates based on the current 3' nucleotide position, and congestion by testing for collisions between polymerases based on a pre-set exclusion radius. An example of a simulation with hard-sphere exclusion is shown in S4 Movie.

5. RNA degradation may in reality be more complex than modeled here, with ribonuclease fluctuations [46], multi-step degradation [47], sequence-dependent degradation [48,49], and transcription-coupled degradation [50] potentially yielding deviations from simple Poisson process degradation. Our simulation-based platform can address these effects analogously to elongation.

Moreover, transcription is, in general, non-stationary due to cell cycle effects [6,16]. Hence, synchronization of data from different cells is important for accurate inference. This may be achieved experimentally by monitoring cues of mitotic state, such as DNA signal or cell shape [6,16].

## Supporting information

**S1 File. Details of the implementation of the algorithm, description of the graphical user interface, and the results of further validation of the search procedure.**
(DOCX)

**S1 Movie. Visualization of transcription dynamics at a single gene copy.**
(MP4)

**S2 Movie. Multi-stage genetic algorithm search over a three-dimensional parameter space.**
(MP4)

**S3 Movie. Visualization of transcription dynamics at two correlated gene copies.**
(MP4)

**S4 Movie. Visualization of transcription dynamics at a single gene copy with hard-sphere exclusion.**
(MP4)

## Acknowledgments

Portions of work by G.G., M.W., and I.G. were performed at Baylor College of Medicine, Houston, Texas, USA. Portions of work by G.G. were performed at Shanghai Jiao Tong University, Shanghai, China. G.G. thanks Dr. Lior Pachter (California Institute of Technology) for support and Dr. Brian Munsky (Colorado State University) for valuable advice. We gratefully acknowledge the computing resources provided by the student innovation center at Shanghai Jiao Tong University.

## Author Contributions

**Conceptualization:** Gennady Gorin, Mengyu Wang, Ido Golding, Heng Xu.

**Formal analysis:** Gennady Gorin, Ido Golding, Heng Xu.

**Funding acquisition:** Gennady Gorin, Mengyu Wang, Ido Golding, Heng Xu.

**Investigation:** Gennady Gorin, Ido Golding, Heng Xu.

**Methodology:** Gennady Gorin, Ido Golding, Heng Xu.

**Project administration:** Ido Golding, Heng Xu.

**Resources:** Heng Xu.

**Software:** Gennady Gorin, Mengyu Wang.

**Supervision:** Ido Golding, Heng Xu.

**Validation:** Gennady Gorin, Ido Golding, Heng Xu.

**Visualization:** Gennady Gorin, Ido Golding, Heng Xu.

**Writing – original draft:** Gennady Gorin, Ido Golding, Heng Xu.

**Writing – review & editing:** Gennady Gorin, Mengyu Wang, Ido Golding, Heng Xu.

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
