## [Decision Letter · Decision Letter 0]

31 Dec 2019

PONE-D-19-31981

Stochastic simulation and statistical inference platform for visualization and estimation of transcriptional kinetics

PLOS ONE

Dear Dr. Xu,

Thank you for submitting your manuscript to PLOS ONE. The paper was sent to two reviewers, who both appreciate the work but raised minor points that I would ask you to adress prior to publication. You can find the reviewers' comments at the bottom of this message.

We would appreciate receiving your revised manuscript by Feb 14 2020 11:59PM. To enhance the reproducibility of your results, we recommend that if applicable you deposit your laboratory protocols in protocols.io, where a protocol can be assigned its own identifier (DOI) such that it can be cited independently in the future. For instructions see: http://journals.plos.org/plosone/s/submission-guidelines#loc-laboratory-protocols

We look forward to receiving your revised manuscript.

Kind regards,

Jordi Garcia-Ojalvo

Academic Editor

PLOS ONE

Journal Requirements:

4. We noted in your submission details that a portion of your manuscript may have been presented or published elsewhere: The manuscript has been released on bioRxiv: https://www.biorxiv.org/content/10.1101/825869v1. The preprint has been uploaded as part of the submission.

Reviewers' comments:

Reviewer's Responses to Questions

**Comments to the Author**

1. Is the manuscript technically sound, and do the data support the conclusions?

Reviewer #1: Yes

Reviewer #2: Yes

2. Has the statistical analysis been performed appropriately and rigorously? 

Reviewer #1: N/A

Reviewer #2: Yes

3. Have the authors made all data underlying the findings in their manuscript fully available?

Reviewer #1: Yes

Reviewer #2: Yes

4. Is the manuscript presented in an intelligible fashion and written in standard English?

Reviewer #1: Yes

Reviewer #2: Yes

5. Review Comments to the Author

Reviewer #1: In this work the authors describe a Matlab platform to perform simulations and rate inference of prokaryotic transcription processes. The stochastic simulations account for promoter on-off switching, initiation, elongation and degradation. An iterative optimization procedure based on a genetic algorithm is implemented to infer the underlying parameters. The manuscript is very clear and everything seems technically correct.

I would only draw attention to the fact that the authors only show the performance of the inference method on a single set of synthetic parameter values with variations on either kon or koff (Fig.2 B-H). I think showing the capability to infer parameters for other sets of parameters, and with modulation of other rates, would much strengthen the work and make it more useful to the community. Similarly, assessing the performance on (published) experimental data and comparing the recovered parameters to those obtained by current techniques based on the random telegraph model would be a relevant contribution.

Reviewer #2: This paper outlines a software toolbox in MATLAB to simulate stochastic dynamics of transcription in prokaryotes. The paper then uses these simulations to infer transcriptional parameters from fluoroscent RNA probe data.

This paper is predicated on the idea that the entire distribution of measurements is important in fitting to a transcriptional model. There are many sources of stochasticity in transcription, even in prokaryotes - promoter state switching, RNA polymerase activity, mRNA degradation, .. in addition to stochasticity due to the readout process by probe hybridization.

This paper does two things - it models all these stochastic aspects as part of a "forward" model, producing putative live-cell and fixed-cell FISH data. The paper then uses the results of this forward model to solve the inverse problem by optimization (i.e., minimizing the output of the forward model and observed experimental data). Their approach to the inverse problem does not assume functional forms for the distributions, which is nice.

I recommend the paper for publication. I ask the authors to clarify the following points to improve the readability of the paper:

Populations vs single cell data - the paper mentions that it concerns itself with both kinds of data. However, the figures and other parts of the text (e.g., early parts of the Discussion) only talk about population-level data. Can the tools described here fit distributions of trajectories (as opposed to distributions at each moment in time)?

What kinds of deviations from the model do you think are most likely during real transcription? E.g., if we don’t find a good fit, do I blame sequence dependence of your rate constants or non-stationarity or something else? Even a short summary of results from the literature on common deviations from the 4 parameter model would be useful here.

The authors simulate millions of cells using Amazon Web Services (AWS) cloud. Do they find that the resulting distributions generally tend to approximated by simple ones common to molecular reactions? If so, can we get by by estimating, e.g. means and variances?

In Fig 2B, why does stage 1 already have a population that covers the target parameters? Is stage 1 shown after some amount of search? If so, it'd be nice to see the initial conditions for the search, to make sure that wasn't chosen to be particularly favorable.

The convergence in Fig 2B appears to go through several "relaxation modes".. At first, there is a quick collapse to a pancake in a particular direction (compare Stage 1 to Stage 2), which then shrinks more slowly. What is the meaning of these 'slow' relaxation directions during the search? There seem to be statements relevant to this collapse in the Methods section (something about a degenerate line for k_{obs}) but it was too cryptic for me to understand. The authors should clarify whether this collapse is an informed choice put in by hand for this particular dataset or if the genetic algorithm naturally collapses the cloud of parameters in this manner. If the former is the case, what guiding principles can a end-user use to figure out which lines to collapse to?

6. PLOS authors have the option to publish the peer review history of their article (what does this mean?). If published, this will include your full peer review and any attached files.

Reviewer #1: No

Reviewer #2: No

---

## [Author Response · Author response to Decision Letter 0]

11 Feb 2020

For detailed responses, please check the attached response letter.

Responses to Reviewer #1

“In this work the authors describe a Matlab platform to perform simulations and rate inference of prokaryotic transcription processes. The stochastic simulations account for promoter on-off switching, initiation, elongation and degradation. An iterative optimization procedure based on a genetic algorithm is implemented to infer the underlying parameters. The manuscript is very clear and everything seems technically correct.

I would only draw attention to the fact that the authors only show the performance of the inference method on a single set of synthetic parameter values with variations on either kon or koff (Fig.2 B-H). I think showing the capability to infer parameters for other sets of parameters, and with modulation of other rates, would much strengthen the work and make it more useful to the community.” 

The reviewer is correct that the performance of our inference method needs to be demonstrated using additional sets of synthetic parameters and with variations of rates other than k_on and k_off. In the previous version of the manuscript, we only fit synthetic data with variations of these parameters due to previous evidence for their role in the modulation of gene expression (Sanchez and Golding 2013). We have now generated more synthetic data with randomly varied parameters (k_on,k_off,k_ini,k_d,v_el). By applying the inference algorithm, we find that the algorithm can infer parameters and reproduce low- and high-order statistics. We show these results in Fig S2, reproduced below. However, even for high-quality synthetic data, convergence to true underlying parameters is not guaranteed, as many sets of parameters can yield the same observables. Such performance corresponds to the degeneration of mapping from the parameter domain to the observable domain, and the inability of the genetic algorithm to report degenerate results from a single search. We suggest that this degeneracy is most practically identified by running the search algorithm multiple times and examining the distribution of the resulting point estimates. High variance in mean estimates over several searches suggests intrinsic non-identifiability, as seen in Fig 2H for k_ini and k_off. 

We describe the above results in the Results section of the main text as well as in the Further Validation section of the Supplementary Information. The relevant text is reproduced below:

In the Results section: For additional validation, we ran the search algorithm using synthetic data generated from random parameter vectors, as well as experimental data from a recent study (6). These procedures are described in the Further Validation section of Supplementary Information. We found that the fits successfully reproduced time-dependent distributions of probe signals. However, agreement between the inferred parameters and ground truth (or, for experimental data, FSP estimates) was not guaranteed, especially for k_ini and k_off. As in Fig 2, these gaps in performance appear to correspond to non-uniqueness in mapping from the parameter domain to the observable domain (29), and inability of the genetic algorithm to report degenerate results. We suggest that this degeneracy is best identified by running the search algorithm multiple times and examining the resulting distribution of point estimates from the centers of the search populations. We take this approach in Fig 2H.

In the Further Validation section of the Supplementary Information: The search algorithm successfully fits the copy-number distributions and mean probe traces. However, its performance in discovering the ground-truth parameter value is fairly poor, especially for k_off and k_ini. This discrepancy essentially speaks to the pervasive degeneracies between the parameter space and the observable space – each observable corresponds to an equivalence class of possible underlying parameter values (1–4).

Figure S2: Validation performance using synthetic data from random parameters. Left column: Comparison of mean probe signal between target and fit (circles: target data, dotted line: mean parameter estimate, shaded region around dotted line: signal spanned by fifty estimates sampled from the one-sigma region). Colors and abscissa as in Fig 2E. Central column: Comparison of copy-number distributions between target and fit (shaded gray regions: target histogram, colored lines: histogram generated from mean parameter estimate, top row/blue: nascent mRNA distribution, bottom row/red: total mRNA distribution). Timepoint values as in Fig 2F. Right column: Final trial parameter population (red: ground truth target, histogram: estimate population, gray line: mean estimate, gray region: one-sigma region of estimates). Variables and limits as in Fig 2C.

“Similarly, assessing the performance on (published) experimental data and comparing the recovered parameters to those obtained by current techniques based on the random telegraph model would be a relevant contribution.”

Following the reviewer’s suggestion, we have demonstrated the performance of our inference method using previously published single-cell experimental data of E. coli transcription (Wang et al. 2019) in the new version of the manuscript. The results are shown in Fig S3, reproduced below. The kinetic parameters of these data were originally extracted using the finite state projection (FSP) technique based on the random telegraph model. The physical model extends previous descriptions (Skinner et al. 2016; Xu et al. 2016; Munsky et al. 2018) by modeling co-transcriptional degradation of mRNA. However, as we state in the Introduction section, solving the chemical master equation (CME) with the full complexity of stochastic stepwise elongation using FSP is technically challenging due to exponential growth in the state space size with increasing resolution. Hence, the FSP method originally used to fit these experimental data relied on a simplified model with a deterministic elongation process (Xu et al. 2016; Wang et al. 2019). In contrast, the inference platform presented in this paper is based on empirical distributions solved from a stochastic simulation of molecular reactions. Therefore, the method can more easily capture the complexities of mRNA production (including stochastic stepwise elongation) and processing. By applying the inference algorithm to the experimental data, we find that, as in the synthetic case, the fits successfully reproduced the target distributions, with performance qualitatively similar to FSP fits in (Wang et al. 2019). However, the parameter values do not agree with those previously derived for FSP. As above, we contend that this occurs due to an intrinsic lack of identifiability.

We describe the above results in the Further Validation section of the Supplementary Information, and the relevant text is reproduced below:

We further analyzed an experimental dataset, previously reported in (5) as a start-up experiment with E. coli grown in a glycerol medium, using the settings given in S3 Table, 100 cells used in stage 1, 10 steps of elongation, and 100 parameter sets. Apart from the pervasive biases at the zero bin, the fits reproduce the target distributions. Qualitatively, the performance is similar to the fits using the FSP algorithm (cf. (5), Supplementary Figure 21). However, the parameter values based on inference from FSP (with additional zero-inflation) are inconsistent with those derived here. This result suggests that the parameter values may not be identified unambiguously, but either method can provide plausible values. A natural next step is the development of extensions to the genetic algorithm to report degenerate results without imposed degeneracy-breaking through recombination.

Figure S3: Validation performance using experimental data. Left column: Comparison of mean probe signal between target and fit (circles: experimental data, dotted line: mean parameter estimate, shaded region around dotted line: signal spanned by fifty estimates sampled from the one-sigma region). Colors and abscissa as in Fig 2E. Central column: Comparison of copy-number distributions between target and fit (shaded gray regions: experimental histogram, colored lines: histogram generated from mean parameter estimate, top row/blue: nascent mRNA distribution, bottom row/red: total mRNA distribution, numbers: minutes since IPTG addition). Right column: Final trial parameter population (green: FSP estimate, histogram: estimate population, gray line: mean estimate, gray region: one-sigma region of estimates). Variables and limits as in Fig 2C.

 

Responses to Reviewer #2

“This paper outlines a software toolbox in MATLAB to simulate stochastic dynamics of transcription in prokaryotes. The paper then uses these simulations to infer transcriptional parameters from fluoroscent RNA probe data.

This paper is predicated on the idea that the entire distribution of measurements is important in fitting to a transcriptional model. There are many sources of stochasticity in transcription, even in prokaryotes - promoter state switching, RNA polymerase activity, mRNA degradation, .. in addition to stochasticity due to the readout process by probe hybridization. 

This paper does two things - it models all these stochastic aspects as part of a "forward" model, producing putative live-cell and fixed-cell FISH data. The paper then uses the results of this forward model to solve the inverse problem by optimization (i.e., minimizing the output of the forward model and observed experimental data). Their approach to the inverse problem does not assume functional forms for the distributions, which is nice.

I recommend the paper for publication. I ask the authors to clarify the following points to improve the readability of the paper:

Populations vs single cell data - the paper mentions that it concerns itself with both kinds of data. However, the figures and other parts of the text (e.g., early parts of the Discussion) only talk about population-level data. Can the tools described here fit distributions of trajectories (as opposed to distributions at each moment in time)? “

We thank the reviewer for pointing out the two strategies of measuring and inferring the single-cell transcriptional kinetics, i.e., the population statistics-based fixed-cell measurement/inference and single trajectory-based live-cell measurement/inference. The former strategy relies on measuring multiple ensembles of snapshot data from fixed-cells at different time points and inferring the transcriptional kinetics of a single cell from time-dependent population statistics (Skinner et al. 2016; Munsky et al. 2018; Wang et al. 2019); while the latter strategy relies on tracking the mRNA signal of live cells over time and directly inferring transcriptional kinetics from single-cell mRNA trajectories (Golding et al. 2005; Larson et al. 2011; Garcia et al. 2013).

Experimentally, the former strategy may be achieved using either smFISH (Femino et al. 1998; Raj et al. 2008) or single-cell RNA-seq (Erhard et al. 2019). Both of these technologies can be applied directly to biological samples without the need for gene modification, and are scalable to the entire transcriptome. In contrast, the latter strategy requires fluorescently labeling mRNA molecules in live cells, which typically relies on genetic modification of the original biological system. Due to the lower signal-to-noise ratio of live imaging data, the technical challenge of applying genetic modification to an arbitrary sample, the possible perturbation of gene activity induced by genetic modification, and the incompatibility with high-throughput measurements, directly measuring the mRNA signal from individual live cells has been less popular than measuring the mRNA signal from a population of fixed cells in previous studies (Specht et al. 2017; George et al. 2018). However, recent advances in live-cell RNA labeling techniques have demonstrated their effectiveness in multiple biological systems (Corrigan et al. 2016; Specht et al. 2017; George et al. 2018; Lammers et al. 2020) and may become more popular in the future.

To infer transcriptional kinetics, the fixed-cell strategy relies on fitting the time-dependent population statistics from several ensembles of snapshot data. With a large cell population, dense time sampling, and a detailed biochemical model, the optimization of distribution divergence can provide a good estimation of transcriptional kinetics (Munsky et al. 2018). Conversely, in the live-cell strategy, the single-cell trajectory data potentially provide additional information about the temporal correlation of transcriptional signals (Larson et al. 2011; Desponds et al. 2016). Hence, with the same amount of data, directly fitting the single-cell trajectory may be ideally more effective and accurate. Practically, achieving the single-cell trajectory fitting is still technically challenging, with few examples in the scientific literature. Specifically, (Tian et al. 2007) fit a single time-series by beginning simulations at each time point, simulating until the next time point, and estimating transition probabilities using a normal likelihood kernel. (Golightly and Wilkinson 2011) fit a single time-series by a Markov Chain Monte Carlo method that used a normal error model to calculate acceptance probabilities. (Daigle et al. 2012) fit a single time-series by simulating ensembles and iteratively selecting parameters that gave trajectories close to observations. (Desponds et al. 2016) used autocorrelation analysis to analyze an occupancy model. (Corrigan et al. 2016) and (Lammers et al. 2020) used likelihood-based hidden Markov models (HMMs) to estimate transition probabilities, and pooled multiple traces by assuming statistical independence. 

The simulation platform that we presented in this manuscript can generate synthetic data for both types of strategies. However, considering that the population statistics-based fixed-cell experiments are more prevalent in the current literature, we only attempted to infer transcriptional kinetics from time-dependent mRNA distributions (as shown in the Results part). To perform the single-cell trajectory-based inference on our non-parametric, non-Bayesian platform, the following challenges need to be considered:

 The simulations may not be initialized with an arbitrary observable, because there exists a large equivalence class of underlying system states that can yield a given probe observation at a particular precision. 

 The observations of autocorrelation are challenging to convert to biophysically interpretable parameters, especially out of steady state.

 Although Bayesian HMM procedures are promising, rigorously recasting them into the context of likelihood-free simulation is problematic. This approach requires either using assuming an error model or calculating likelihoods from a kernel. The choice of kernel is unclear, and potentially fraught with challenges for multimodal data. 

As a conceptual inspiration from Daigle et al. (Daigle et al. 2012), we suggest that our simulation-based platform may be used for single-cell trajectory fitting in the following way. For a small live-cell dataset of N cells and n timepoints, it is possible to initialize N genetic algorithm searches that iterate over n stages to incrementally shrink the plausible parameter space to values consistent with the time-series, as described in Methods. Afterward, the independent searches may be combined to find a single common plausible parameter region. Qualitatively, at each step, this approach finds a region with a high probability of achieving a transition between values at two time points, then conditions on it for the consequent transition. The recombination is analogous to the pooling of multiple traces. 

The implementation of this algorithm is outside of the scope of the manuscript. We consider the derivation and implementation of time-series fitting methods a valuable direction for future versions of the platform. In particular, we anticipate it may bring computational advantages over current methods. For example, the recent publication (Lammers et al. 2020) mentions that the HMM analysis of 25 multi-trace datasets takes approximately two hours on 24 CPU cores. Seven out of the ten searches in Fig S2 took under 25 minutes on 8 CPU cores; the others took multiple hours, but this computational load may be mitigated using adaptive methods, as well as parallelization across more cores. Therefore, considering the computational expense of the HMM framework, we anticipate a simulation-based approach presents a viable alternative.

On the whole, we summarize this part in the Discussion section and the relevant text is reproduced below:

The parameter estimation procedure only uses time-dependent histograms: the platform can generate live- and fixed-cell data, but only attempts to fit fixed-cell data. These biochemical distinctions induce methodological differences for parameter inference. Fixed-cell measurements are necessarily destructive, and kinetics may only be inferred from distribution-level data. In contrast, live-cell signals contain additional information regarding the temporal correlation of a given cell. In the current study, we focus on fitting distribution data for two reasons. Firstly, inference from ensembles can be directly implemented using a variety of divergence metrics that make minimal assumptions regarding the form of the data (25). On the other hand, inference from time-series requires error models for transitions between observed states, which are generally intractable (34). Secondly, fixed-cell measurements are amenable to high-throughput experiments, can be scaled to the entire transcriptome via multiplexing (35), produce better signal/noise behavior, and do not require genetic modification (36), contributing to their greater popularity (36). Therefore, we have optimized the parameter estimation method for the most likely current use case of inference from fixed-cell experiments.

Recent advances in live-cell labeling techniques do suggest that the method may become more practical and popular in the future (37,38). To anticipate this, we propose several approaches to live-cell data inference, motivated by previous efforts. If the dataset is large enough, the fixed-cell procedure may be sufficient, discarding the temporal correlation information altogether (25). Alternatively, it is possible to iterate through the data points of a time-series, generating an ensemble of transitions, estimating the likelihood of the observed transition based on a kernel, and optimizing the likelihood by varying model parameters. This approach has been useful for relatively small datasets (34,39,40). However, its application to multimodal time-series is potentially problematic due to the assumption of smoothness, the complexity of developing robust adaptive kernels, and the well-documented problems accompanying kernel density estimation of multivariate data (41). Further, it presents computational challenges: the different increments are ostensibly independent due to the Markov property, but the non-unique mapping from the underlying Markov states to the observed probe data prevents the independent initialization of each increment. This feature makes it infeasible to parallelize the estimation of transition probabilities over non-overlapping increments. Several recent publications perform likelihood-based inference on hidden Markov models (37,38). However, rigorously recasting these methods into the context of likelihood-free simulation is challenging, as is their extension to multimodal data. We suggest that the algorithm described in the Methods section can be extended to treat time-series data. Such an algorithm may iterate over a single time-series to incrementally shrink to a consistent parameter region. The selection of the region is based on a non-parametric error metric between the target fluorescence and the ensemble distribution for each trial parameter at the end of each interval. Conceptually, this process iteratively identifies parameter values by optimizing for observed transitions, analogously to previous work (40). Afterward, independent searches over multiple traces may be aggregated to find a single plausible region. Given the computational expense of current HMM-based methods (38), an adaptive simulation-based approach may present a viable alternative.

“What kinds of deviations from the model do you think are most likely during real transcription? E.g., if we don’t find a good fit, do I blame sequence dependence of your rate constants or non-stationarity or something else? Even a short summary of results from the literature on common deviations from the 4 parameter model would be useful here. “

The reviewer is correct that multiple factors may affect the accuracy of our model when applying to real transcription. We now summarize these factors in the Discussion section and the relevant text is reproduced below:

Our platform models the activity of individual gene loci in non-compartmentalized prokaryotic cells with the assumption that transcription follows a two-state random telegraph model with time-homogeneous rate parameters, and elongation and degradation are described by multistep Poisson processes. These assumptions may be violated in the following ways:

 The description of a eukaryotic system may be of interest. The implementation of eukaryotic transcription would require making significant changes to the reaction schema, such as disabling the degradation of nuclear mRNA and adding a kinetic model of a transport process after the release of the newly transcribed mRNA. 

 Multiple gene copies may be present in a cell (6). It is straightforward to extend the current model to account for this physiology. For example, S3 Movie shows the correlated dynamics at two gene copies, which may only turn on when an underlying Boolean cell state is on.

 The two-state switching of gene activation/inactivation may be an over-simplified picture of gene activity. In reality, an N-state model may be more accurate (15,42,43). To consider this effect, our simulation-based framework can be easily extended to include more gene states rather than a single Boolean state.

 The transcription elongation rate may not be constant, whether due to sequence dependence (44) or polymerase congestion (13,14,45). The implementation of these rules is challenging using the CME framework. Our simulation-based platform can incorporate sequence-dependent rates by adjusting rates based on the current 3' nucleotide position, and congestion by testing for collisions between polymerases based on a pre-set exclusion radius. An example of a simulation with hard-sphere exclusion is shown in S4 Movie. 

 RNA degradation may in reality be more complex than modeled here, with ribonuclease fluctuations (46), multi-step degradation (47), sequence-dependent degradation (48,49), and transcription-coupled degradation (50) potentially yielding deviations from simple Poisson process degradation. Our simulation-based platform can address these effects analogously to elongation.

Moreover, transcription is, in general, non-stationary due to cell cycle effects (6,16). Hence, synchronization of data from different cells is important for accurate inference. This may be achieved experimentally by monitoring cues of mitotic state, such as DNA signal or cell shape (6,16).

“The authors simulate millions of cells using Amazon Web Services (AWS) cloud. Do they find that the resulting distributions generally tend to approximated by simple ones common to molecular reactions? If so, can we get by by estimating, e.g. means and variances?”

Based on previous studies and observations of our simulation, the shapes of mRNA distribution of random telegraph models may be classified into several groups (monomodal, bimodal, etc. (Munsky et al. 2012; Xu et al. 2016)). Yet the distributions don't tend to approach elementary function forms in any practical way once the submolecular probe features are incorporated (Xu et al. 2016). Hence, a quantitative approximation of distributions using simple function forms requires further validation in theory, which, to date, is still lacking. 

Particularly, the reviewer is correct that moments of mRNA distributions, such as means and variances, are functions of kinetic parameters. For example, in our platform, the mean of the observed probe signal is analytically calculated from the kinetic parameters (i.e., via quadrature rather than via FSP). However, estimating transcriptional kinetics from these moments (or other similar quantities) may not be easier than fitting the entire distribution for the following reasons:

 Each moment corresponds to an equivalence class of parameters, i.e., many combinations of kinetic parameters can give rise to the same moment. To narrow the parameter space, multiple moments need to be considered simultaneously. Specifically, our model requires fitting five moments in order to estimate the model’s five free parameters. However, we are unaware of any easily tractable expressions for the higher moments of a multi-state gene with arbitrary fluorescent probe coverage. Deriving the mathematical expressions of these moments may not be simpler than fitting the entire distribution. 

 Even if the expressions of higher moments can be derived, optimizing the estimates would require an error model, which is not available in the regime of low copy numbers. In comparison, we limit this issue in our inference algorithm with search stages that use the entire distribution.

We summarize the above points in the Discussion section, and the relevant text is reproduced below: 

On the other hand, we suggest that five-parameter inference entirely from moments is infeasible at this time. Typically, fitting n parameters requires n moments. For the current system, signal expectations can be computed (6), but expressions for the higher moments are unknown. Even if they were available, the choice of error model for these higher moments is far from clear, especially in the physiologically important regime of low copy numbers. Furthermore, we anticipate that the value of this heuristic method rests in applications to models with ad hoc mechanisms whose physics are challenging to approach analytically.

“In Fig 2B, why does stage 1 already have a population that covers the target parameters? Is stage 1 shown after some amount of search? If so, it'd be nice to see the initial conditions for the search, to make sure that wasn't chosen to be particularly favorable.”

We appreciate this feedback and opportunity to clarify the procedure. The initial condition for the search is drawn from a log-uniform distribution across the entire search space. We describe this point in the Methods section as follows:

The parameter domain is shown in Figure 2C. We initialize the search using a uniform distribution over the full parameter domain.

The distribution for Stage 1 is the result of the stage. We describe this point in the caption to Figure 2 as follows:

B: Convergence of the genetic algorithm at the end of each stage of the search (red: ground truth target, gray: population of parameter estimates).

“The convergence in Fig 2B appears to go through several "relaxation modes".. At first, there is a quick collapse to a pancake in a particular direction (compare Stage 1 to Stage 2), which then shrinks more slowly. What is the meaning of these 'slow' relaxation directions during the search? There seem to be statements relevant to this collapse in the Methods section (something about a degenerate line for k_{obs}) but it was too cryptic for me to understand. The authors should clarify whether this collapse is an informed choice put in by hand for this particular dataset or if the genetic algorithm naturally collapses the cloud of parameters in this manner. If the former is the case, what guiding principles can a end-user use to figure out which lines to collapse to?”

We thank the reviewer for raising the question about different relaxation modes. The "slow" relaxation directions correspond to observables that are weak functions of the variable under examination, while the "fast" ones correspond to observables that are strong functions of the variable. 

As a simple example, the mean total amount of mRNA is a strong function of the degradation rate. The simplest ODE model for the total amount of RNA is dT/dt=k_i^obs-k_d T. The turn-on initial condition T(t=0)=0 yields the solution (k_i^obs)/k_d (1-e^(-k_d ) ). Stage 2 of the search fits the mean of the total amount of RNA and yields sharp estimates for (k_i^obs)/k_d and k_d. Analogously, stage 4, which fits the mean nascent mRNA signal, yields sharp estimates for v_el.

This idea is virtually identical to the Fisher information in Bayesian parametric statistics. However, it is challenging to formalize this in a simulation-based, likelihood-free context, so we suggest using this analogy with caution.

The order of collapse is user-determined through the order of optimization stages; however, for each stage, the direction of collapse is guided by the information content with respect to each parameter. For relatively simple models, basic physical insight is sufficient to draw connections between observables and parameters, e.g., via simplified ODE representations that abstract away gene dynamics and submolecular features. For more complex models, exploratory analysis is necessary.

We summarize the above points in the Discussion section, and the relevant text is reproduced below:

Even without moment-based analytical constraints, it is possible to use physical considerations to guide the development of optimization metrics. For example, in a Bayesian framework, the Fisher information of the mean total probe signal is high with respect to k_d, but low with respect to v_el. As shown in Fig 2B, stage 2, which optimizes the total mean probe signal, provides a tight bound on k_d but not v_el; conversely, stage 4, which optimizes the mean nascent probe signal, yields a tight bound on v_el. For more complex models, exploratory analysis is necessary to determine the coupling between observables and parameters, but the provided heuristics and physical expectations provide a starting point.

 

References

Corrigan AM, Tunnacliffe E, Cannon D, Chubb JR. A continuum model of transcriptional bursting. eLife. 2016 Feb 20;5:e13051. 

Daigle BJ, Roh MK, Petzold LR, Niemi J. Accelerated maximum likelihood parameter estimation for stochastic biochemical systems. BMC Bioinformatics. 2012 Dec;13(1):68. 

Desponds J, Tran H, Ferraro T, Lucas T, Perez Romero C, Guillou A, et al. Precision of Readout at the hunchback Gene: Analyzing Short Transcription Time Traces in Living Fly Embryos. PLoS Comput Biol. 2016 Dec 12;12(12):e1005256. 

Erhard F, Baptista MAP, Krammer T, Hennig T, Lange M, Arampatzi P, et al. scSLAM-seq reveals core features of transcription dynamics in single cells. Nature. 2019 Jul;571(7765):419–23. 

Femino AM, Fay FS, Fogarty K, Singer RH. Visualization of Single RNA Transcripts in Situ. 1998;280:7. 

Garcia HG, Tikhonov M, Lin A, Gregor T. Quantitative Imaging of Transcription in Living Drosophila Embryos Links Polymerase Activity to Patterning. Current Biology. 2013 Nov;23(21):2140–5. 

George L, Indig FE, Abdelmohsen K, Gorospe M. Intracellular RNA-tracking methods. Open Biol. 2018 Oct;8(10):180104. 

Golding I, Paulsson J, Zawilski SM, Cox EC. Real-Time Kinetics of Gene Activity in Individual Bacteria. Cell. 2005 Dec;123(6):1025–36. 

Golightly A, Wilkinson DJ. Bayesian parameter inference for stochastic biochemical network models using particle Markov chain Monte Carlo. Interface Focus. 2011 Dec 6;1(6):807–20. 

Lammers NC, Galstyan V, Reimer A, Medin SA, Wiggins CH, Garcia HG. Multimodal transcriptional control of pattern formation in embryonic development. PNAS. 2020;117(2):836–47. 

Larson DR, Zenklusen D, Wu B, Chao JA, Singer RH. Real-Time Observation of Transcription Initiation and Elongation on an Endogenous Yeast Gene. Science. 2011 Apr 22;332(6028):475–8. 

Munsky B, Li G, Fox ZR, Shepherd DP, Neuert G. Distribution shapes govern the discovery of predictive models for gene regulation. Proc Natl Acad Sci USA. 2018;115(29):7533–8. 

Munsky B, Neuert G, van Oudenaarden A. Using Gene Expression Noise to Understand Gene Regulation. Science. 2012;336(6078):183–7. 

Raj A, van den Bogaard P, Rifkin SA, van Oudenaarden A, Tyagi S. Imaging individual mRNA molecules using multiple singly labeled probes. Nat Methods. 2008 Oct;5(10):877–9. 

Sanchez A, Golding I. Genetic Determinants and Cellular Constraints in Noisy Gene Expression. Science. 2013 Dec 6;342(6163):1188–93. 

Skinner SO, Xu H, Nagarkar-Jaiswal S, Freire PR, Zwaka TP, Golding I. Single-cell analysis of transcription kinetics across the cell cycle. eLife. 2016 Jan 29;5:e12175. 

Specht EA, Braselmann E, Palmer AE. A Critical and Comparative Review of Fluorescent Tools for Live-Cell Imaging. Annu Rev Physiol. 2017 Feb 10;79(1):93–117. 

Tian T, Xu S, Gao J, Burrage K. Simulated maximum likelihood method for estimating kinetic rates in gene expression. Bioinformatics. 2007 Jan 1;23(1):84–91. 

Wang M, Zhang J, Xu H, Golding I. Measuring transcription at a single gene copy reveals hidden drivers of bacterial individuality. Nat Microbiol. 2019 Sep 16;4:2118–27. 

Xu H, Skinner SO, Sokac AM, Golding I. Stochastic Kinetics of Nascent RNA. Phys Rev Lett. 2016;117(12):128101.

---

## [Decision Letter · Decision Letter 1]

9 Mar 2020

Stochastic simulation and statistical inference platform for visualization and estimation of transcriptional kinetics

PONE-D-19-31981R1

Dear Dr. Xu,

We are pleased to inform you that your manuscript has been judged scientifically suitable for publication and will be formally accepted for publication once it complies with all outstanding technical requirements.

With kind regards,

Jordi Garcia-Ojalvo

Academic Editor

PLOS ONE

Additional Editor Comments (optional):

Reviewers' comments:

Reviewer's Responses to Questions

**Comments to the Author**

1. If the authors have adequately addressed your comments raised in a previous round of review and you feel that this manuscript is now acceptable for publication, you may indicate that here to bypass the “Comments to the Author” section, enter your conflict of interest statement in the “Confidential to Editor” section, and submit your "Accept" recommendation.

Reviewer #1: All comments have been addressed

Reviewer #2: All comments have been addressed

2. Is the manuscript technically sound, and do the data support the conclusions?

Reviewer #1: Yes

Reviewer #2: Yes

3. Has the statistical analysis been performed appropriately and rigorously? 

Reviewer #1: N/A

Reviewer #2: Yes

4. Have the authors made all data underlying the findings in their manuscript fully available?

Reviewer #1: Yes

Reviewer #2: Yes

5. Is the manuscript presented in an intelligible fashion and written in standard English?

Reviewer #1: Yes

Reviewer #2: Yes

6. Review Comments to the Author

Reviewer #1: (No Response)

Reviewer #2: The authors have sufficiently addressed the issues I raised. The most pressing issue I raised was showing the performance on additional synthetic data, instead of relying on one particular set. I appreciate the authors doing so and plainly reporting the degeneracy in going from ground truth parameters to observables. They explicitly clarify that their algorithm should be run multiple times to understand such degeneracies. They have also added applications to other real datasets.

7. PLOS authors have the option to publish the peer review history of their article (what does this mean?). If published, this will include your full peer review and any attached files.

Reviewer #1: Yes: Rosa Martinez-Corral

Reviewer #2: No

---

## [Editor Report · Acceptance letter]

13 Mar 2020

PONE-D-19-31981R1 

Stochastic simulation and statistical inference platform for visualization and estimation of transcriptional kinetics 

Dear Dr. Xu:

I am pleased to inform you that your manuscript has been deemed suitable for publication in PLOS ONE. Congratulations! Your manuscript is now with our production department. 

With kind regards,

on behalf of

Dr. Jordi Garcia-Ojalvo 

Academic Editor

PLOS ONE